# The Utility of Video Recording in Assessing Bariatric Surgery Complications

**DOI:** 10.3390/jcm11195573

**Published:** 2022-09-22

**Authors:** Marius Nedelcu, Sergio Carandina, Patrick Noel, Henry-Alexis Mercoli, Marc Danan, Viola Zulian, Anamaria Nedelcu, Ramon Vilallonga

**Affiliations:** 1ELSAN, Clinique Saint Michel, Centre Chirurgical de l’Obesite, 83000 Toulon, France; 2ELSAN, Clinique Bouchard, 13000 Marseille, France; 3Emirates Specialty Hospital, Dubai 505240, United Arab Emirates; 4ELSAN, Polyclinique de Franche-Comté, 25000 Besançon, France; 5Universitat Autònoma de Barcelona, 08028 Barcelona, Spain; 6Endocrine, Metabolic and Bariatric Unit, General Surgery Department, Hospital Vall d’Hebron, 08023 Barcelona, Spain

**Keywords:** sleeve, complications, procedure video recording, learning curve, reintervention

## Abstract

Introduction: Recording every procedure could diminish the postoperative complication rates in bariatric surgery. The aim of our study was to evaluate the correlation between recording every bariatric surgery and their postoperative analysis in relation to the early or late postoperative complications. Methods: Seven hundred fifteen patients who underwent a bariatric procedure between January 2018 and December 2019 were included in a retrospective analysis. There were: 589 laparoscopic sleeve gastrectomies (LSGs); 110 Roux-en-Y bypasses (RYGBs) and 16 gastric bands (LAGBs). The video recording was systematically used, and all patients were enrolled in the IFSO registry. Results: There were 15 patients (2.1%) with surgical postoperative complications: 5 leaks, 8 hemorrhages and 2 stenosis. Most complications were consequent to LSG, except for two, which occurred after RYGB. In four cases a site of active bleeding was identified. After reviewing the video, in three cases the site was correlated with an event which occurred during the initial procedure. Three out of five cases of leak following sleeve were treated purely endoscopically, and no potential correlated mechanism was identified. Two other possible benefits were observed: a better evaluation of the gastric pouch for the treatment of the ulcer post bypass and the review of one per operative incident. Two negative diagnostic laparoscopies were performed. The benefit of the systematic video recording was singled out in eight cases. All the other cases were completed by laparoscopy with no conversion. Conclusion: To record every bariatric procedure could help in understanding the mechanism of certain complications, especially when the analysis is performed within the team. Still, recording the procedure did not prevent the negative diagnostic laparoscopy, but it could play a significant role for the medico-legal aspect in the future.

## 1. Introduction

The increasing prevalence of morbid obesity and the fact that surgery is the only effective treatment have led to an increased number of bariatric procedures. For the last decade, the number of bariatric procedures has constantly increased, and the laparoscopic sleeve gastrectomy (LSG) has become the most frequent bariatric procedure both in France and worldwide [1,2], while the number of Roux-en-Y bypasses (RYGBs) remains steady. Although LSG does not require a reconstructive time with the creation of gastrointestinal anastomoses, it nevertheless presents certain fundamental surgical steps that need a certain level of experience in laparoscopic and bariatric surgery to reduce the risk of postoperative complications [3]. Several publications have advocated the critical role of the learning curve for different bariatric procedures as it relates to diminishment of postoperative complication rates [4,5]. Even if many surgeons consider LSG to be a very simple or technically easy procedure, there are limited data [6] in the literature regarding the experience needed in order to significantly reduce the risk of gastric leak (GL).

Even if the GL prevalence has decreased steadily and recent series [7,8,9] have reported a rate of 1 to 2%, the interest to identify potential mechanisms of leaks remains significant/substantial in the literature. GL is the most serious complication of this procedure due to multiple alterations influencing wound healing near the gastroesophageal junction, such as increased pressure, stricture formation and a too-narrow sleeve, mismatched staple height and tissue thickness, vascular supply and energy sources, among others. Considering all these factors, it is essential to keep any possible information regarding the procedure. Furthermore, recording every procedure could also reduce the learning curve in bariatric surgery as it has been proven for basic laparoscopic skills [10].

The aim of our study was to evaluate the interest of recording every bariatric procedure and to analyze postoperatively both the early and late postoperative complications.

## 2. Materials and Methods

All patients with postoperative complications after different bariatric procedures, from January 2018 to December 2019, were selected for the analysis of their video recording for both the initial surgery and the reintervention. All procedures were performed by two surgeons (M.N. and S.C.). The study included all consecutive patients who underwent different bariatric procedures in this period. No patients were excluded from the study, and all patients were included in the IFSO (International Federation Of Societies of Obesity) register. There were included data on patient demographic characteristics, case history, operative details, complications and one-year results.

All postoperative complications were selected, and all video recordings related to these patients were analyzed by two bariatric surgeons (M.D. and P.N.) with more than 4000 bariatric procedure for each.

For patients with bleeding, an identification of the origin of the bleeding was attempted. Equally, the control of blood pressure during stapling, at the end of the procedure and in the postoperative period was analyzed. For leak following LSG, several key steps were analyzed:Respecting the angulus with the tip of the first stapling;Distance of stapling from the Hiss angle;Crossing the staple lines;Extensive use of energy device;Final form of the sleeve.

All procedures performed in studies involving human participants were in accordance with the ethical standards of the institutional and/or national research committee and with the 1964 Helsinki declaration and its later amendments or comparable ethical standards. For this type of study, formal consent is not required.

### Operatory Technique

The posterior approach with the 3-port technique remained constant from the beginning of the experience, and it has previously been described [11]. Once the stomach has been freely dissected, it is then transected over a 37-Fr rigid calibration tube (MidSleeve, MID, Dardilly, France). All patients were followed up on an outpatient basis regularly over the entire period. The follow-up consisted in a careful documentation of changes in weight, comorbidities and any reported complication. Video analysis was performed systematically whenever a patient presented in the postoperative period with tachycardia, unsuspected pain. Standard management was performed, including CT-San and blood sample test. After primary suspected diagnosis, video recording was conducted in order to increase data and information regarding the procedure. For patients with stenosis following LGS in the postoperative period or in the mid-long term, video recording was also reviewed.

## 3. Results

A total of 715 bariatric procedures performed between January 2018 and December 2019 were analyzed. There were: 589 laparoscopic sleeve gastrectomies (LSGs); 110 Roux-en-Y bypasses (RYGBs) and 16 gastric bands (LAGBs). The complications were classified according to the Dindo–Clavien scale. For the current study, we included only 15 patients (2.1%) with major complications (Clavien ≥ 3) in order to evaluate the benefits of video recording. Considering each procedure there were:Thirteen cases (2.2%) following LSG: five leaks (0.8%), seven bleeding and one stenosis.Two cases following RYGB: one bleeding from jejuno-jejunal anastomosis and one stenosis of gastro-esophageal anastomosis.No cases following LAGB.

In four out of seven cases of bleeding following LSG, an active site of bleeding was identified. All patients were reoperated, and in cases with active bleeding, the hemostasis was achieved by suturing the staple line (figure of X). The evacuation of the hematoma and drainage were systematically performed. After reviewing the video, in four cases the site was correlated with the initial procedure. Upon analyzing the blood pressure at the end of the procedure and in the postoperative period for three cases, a difference of more than 50 mmHg for systolic blood pressure was observed and concluded as the main factor for the occurrence of bleeding.

Concerning the GL following LSG, in three out of five cases the treatment was purely endoscopic by the insertion of repetitive pigtails and one case of endoscopic septotomy with balloon dilatation. In all these three cases the initial procedure was reviewed, but in the absence of surgical re-exploration, no potential mechanism of GL was identified. For the other two cases, the potential mechanisms of leak were identified as: use of an inappropriate size of staples and crossing of the staple line. For these five cases of leaks, the operative time was analyzed with a mean of 42 min (range 28–54). The stenosis was explained as misalignment between the first and the second staplers, and the helical stenosis was confirmed and simultaneously treated by endoscopic dilatation.

Regarding the RYGB, the two complications were explained after video analysis by different factors. The stenosis of the gastro-jejunal anastomosis was explained by two factors: initial experience with manual anastomosis and patient with important smoking habits. The treatment consisted in repetitive endoscopic dilatations followed by deployment of a short endoscopic stent for 4 weeks with complete resolution. The hemorrhage of jejuno-jejunal anastomosis was also explained by inappropriate use of the staple heights. One day after the initial procedure, a laparoscopic exploration was performed with complete running suture of the anastomosis.

The video recording can offer two other potential benefits: a better evaluation of the gastric pouch for the treatment of ulcer postbypass and the review of one peroperative incident: stapling the tube during RYGB. Two negative diagnostic laparoscopies were performed, and the benefit of systematic video recording was identified in eight cases (53.3%).

All other cases were completed by laparoscopy with no conversion to open surgery or no mortality recorded.

## 4. Discussion

LSG has increasingly gained worldwide acceptance among bariatric surgeons during the last 10 years, becoming the most common procedure. For LSG we can consider two learning curves. The first one is common with other laparoscopic procedures, and it concerns the ability to perform an LSG. The second one, more important, is related to significantly decreasing the risk of a complication, namely the GL. Leaks are estimated to be the most serious complication of LSG for at least two reasons. From an observational point of view, an LSG leak appears to pose more difficulty in healing compared with other leaks in bariatric surgery. This could probably be explained by the mechanism itself of the LSG which is the creation of a high-pressure gastric tube often associated with a functional angular stenosis [12]. The association of stenosis and leaks has been documented, and it should be treated simultaneously; ignorance of a stricture may prolong the duration of a leak [13]. The second reason lies in the lack of standardization in the management of the fistula, a continuously evolution of the endoscopic approach. The topic of the leak following LSG has been largely discussed in the literature during the last decade. Throughout this period, the rate of chronic fistula has had a significant decrease due to several factors: improved surgical experience, enhanced quality of staplers and avoiding smoking habits for patients. Our team reported the importance in the learning curve to diminish 10-fold the risk of leak in a previous manuscript [7].

Varban et al. [14] have analyzed technical variation among surgeons (n = 30) who voluntarily submitted a video of a typical LSG. They have reported that surgical complication rates ranged from 0 to 4.32% and concluded that top ranked surgeons did have faster mean operative times, indicating that there may be other metrics of technical quality that correlate to optimal outcomes. Unfortunately, in our study, the operative time was not evaluated for all the procedures, but only for the five cases of leaks with a mean of 42 min (range 28–54). This cannot be considered as a risk factor for leak occurrence. Certainly, the operative time should be investigated more in prospective studies.

The future belongs to artificial intelligence (AI), as it has been reported by Hasimoto et al. [15]. They have developed AI algorithms to identify operative steps in LSG. These steps should be revised, since liver retraction, liver biopsy or bagging specimen cannot be considered through the seven main operative steps of LSG. Based on our previous experience with complications following bariatric procedures [7,12,13], in our current manuscript we have introduced others important operative steps such as: respect of the incisura angularis, staple line alignment or sectioning at the level of the cardio-esophageal junction as important steps in preventing major complications following LSG.

Stress has been shown to impact adversely multiple facets critical to optimal performance [16]. Advancements in wearable technology can reduce barriers to observing stress during surgery. Recording every surgical procedure has also some limitations such as not being able to evaluate the surgical stress during the bariatric procedure. Grantcharov et al. [17] have analyzed the risk to cause an injury to the patient or posed a risk of harm. They have reported higher rates of events (47–66% higher) in the higher stress quantiles than in the lower stress quantiles of acute mental stress. They have concluded that there is an association between measures of acute mental stress and worse technical surgical performance. Even if our study did not beneficiate of this accurate evaluation, we can still mention that two of the five leaks identified occurred when the surgeon was under stress from the evolution of another patient.

Van Dalen et al. [18] reported also that surgical safety may be improved using a medical data recorder for the purpose of postoperative team debriefing. It provides the team in the operating room with the opportunity to look back upon their joint performance objectively, to discuss and learn from suboptimal situations or possible adverse events. Comparing to the aviation black box, recording as much information from the operative time will certainly play a role for the insurance company as well. Our study results present two main limitations. First, all the procedures were performed only by two surgeons, and the effect of the learning curve could not be evaluated as Varban et al. did for 30 surgeons [14]. Secondly, in our study, the operative time was not evaluated for all the procedures, and it could have led to a better understanding postoperative complications. In our current daily activity, the video recording is used mainly in case of complication. All the procedures are systematically recorded, and only in a case of a potential complication is the primary video reviewed to identify the complication. After the reintervention, the second video will be analyzed by multiple surgeons with the intent to identify the potential mechanism of the complication.

## 5. Conclusions

Systematic procedure video recording facilitates feedback reflection and self-directed learning, which improves the ability to understand the mechanisms of different complications in bariatric surgery. Combining both self-assessment and video feedback may be beneficial due to the advantages of video review.

## Data Availability

The data presented in this study are available on request from the corresponding author.

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
