# Peer review of "The Utility of Video Recording in Assessing Bariatric Surgery Complications"

_jcm, 2022, doi:10.3390/jcm11195573_

Round 1

Reviewer 1 Report

The authors present a timely and intriguing study on the correlation between bariatric surgery video recording and postoperative complications. The bariatric literature is beginning to see more studies evaluating the quality and/or utility of surgical videos not only for education and simulation, but also for ensuring optimal surgical outcomes, as this study describes.

With major revisions, the present study may be accepted for publication. First, in the Methods, rather than giving the years of experience of each video reviewer, please give the number of LSG and RYGB (>1000? >2000?) done by each video reviewer. This is more in line with similar studies on evaluating surgical videos. 

The major revision needed is the addition of quantitative methods to justify a correlation between surgical video evaluation and postoperative patient course. Perhaps an ANOVA comparing lengths of stay, readmits, early postoperative complications of patients whose surgical videos were reviewed vs. those who were not?

Author Response

We thank the reviewers for their fair and very constructive feedback. We have done the appropriate modifications according to our experience and convictions. We are convinced that by the modifications done to the manuscript according to your suggestions we have highly improved the quality of our paper.

The authors present a timely and intriguing study on the correlation between bariatric surgery video recording and postoperative complications. The bariatric literature is beginning to see more studies evaluating the quality and/or utility of surgical videos not only for education and simulation, but also for ensuring optimal surgical outcomes, as this study describes.

Thank you very much for your positive feedback, this was exactly our main endpoint and we are convinced that the current manuscript will stimulate other teams to report their experience. Also, in the future probably the recording in the operating room will become mandatory for legal purposes as there is the black box for any flight.

With major revisions, the present study may be accepted for publication. First, in the Methods, rather than giving the years of experience of each video reviewer, please give the number of LSG and RYGB (>1000? >2000?) done by each video reviewer. This is more in line with similar studies on evaluating surgical videos. 

Thank you very much for your suggestion, the manuscript was modified accordingly.

The major revision needed is the addition of quantitative methods to justify a correlation between surgical video evaluation and postoperative patient course. Perhaps an ANOVA comparing lengths of stay, readmits, early postoperative complications of patients whose surgical videos were reviewed vs. those who were not?

Any comparation with a control group of non-recorded procedure would not be possible as the video recording was systematically used and this was mentioned both in the abstract and several times throughout the manuscript. After responding to the other reviewer it will be difficult to do an analysis and to prove the role for the learning curve. Accordingly the manuscript was revised to evaluate the utility of video recording in assessing post bariatric surgery complications and not the learning curve.

We thank the reviewers for their fair and very constructive feedback. We have done the appropriate modifications according to our experience and convictions. We are convinced that by the modifications done to the manuscript according to your suggestions we have highly improved the quality of our paper.

Reviewer 2 Report

I would like to thank the editors for sending me this paper to review.

I found the topic very interesting. However, I did have several serious concerns.

1-The titel does not represent the topic of the paper. Better suggested titles eg (usefufulness of video recording in assessing post bariatric surgery complications).

2-You focused throughout the article that video recordings enhances the learning curve. Your methodology however focused on a completely differernt thing. There was nothing mentioning how learning curve is assessed. And you based the whole discussion on that rather than on your results outcome.

I think that the methods and the results are clearly presented but I think you need to reformat the discussion and the whole paper in a away to focus on your results. for example present the rates, what were they were caused by and how to avoid them and if you add your own recommendation on how the video recording can have a positive impact in the field and not the whole discussion on it since this is not what the paper is about (according to your methodology).

What is your recommendations  regarding video taping and reviewing them and who should do them. Do I review the tape after the surgery immediately, or after a complication, or before every patient gets admitted with an emergency. Are you suggesting video taping as a diagnostic tool? for emergency unwell patients. Please clarify. So just add what your thoughts about the topic are at end of discussion but more focus on methodology topic.

Author Response

We thank the reviewers for their fair and very constructive feedback. We have done the appropriate modifications according to our experience and convictions. We are convinced that by the modifications done to the manuscript according to your suggestions we have highly improved the quality of our paper.

I would like to thank the editors for sending me this paper to review.

I found the topic very interesting. However, I did have several serious concerns.

Thank you very much for your positive feedback.

1-The title does not represent the topic of the paper. Better suggested titles eg (usefufulness of video recording in assessing post bariatric surgery complications).

Thank you very much for your comment. As you suggested, the titled was modified to “The utility of video recording in assessing post bariatric surgery complications”

2-You focused throughout the article that video recordings enhances the learning curve. Your methodology however focused on a completely differernt thing. There was nothing mentioning how learning curve is assessed. And you based the whole discussion on that rather than on your results outcome.

I think that the methods and the results are clearly presented but I think you need to reformat the discussion and the whole paper in a way to focus on your results. for example present the rates, what were they were caused by and how to avoid them and if you add your own recommendation on how the video recording can have a positive impact in the field and not the whole discussion on it since this is not what the paper is about (according to your methodology).

Thank you very much for your recommendation. The new revised form of the manuscript is focused more on the utility of video recording and less on the learning curve. We are completely agree with you that proving that systematic video recording influences the learning curve will be very difficult, even if it is obvious that looking to our mistakes will improve our future outcomes.

What is your recommendations  regarding video taping and reviewing them and who should do them. Do I review the tape after the surgery immediately, or after a complication, or before every patient gets admitted with an emergency. Are you suggesting video taping as a diagnostic tool? for emergency unwell patients. Please clarify. So just add what your thoughts about the topic are at end of discussion but more focus on methodology topic.

Thank you very much for your comment, it is difficult to use the term “our recommendations” as I consider that any “recommendation” should come from a medical society, we have only personal experience. The following paragraph was added to the revised form of the paragraph: “In our current daily activity, the video recording is used mainly in case of complication. All the procedures are systematically recorded, and only in a case of a potential complication, the primary video is reviewed to identify the complication. After the reintervention, the second video will be analyzed by multiple surgeons with the intent to identify the potential mechanism of the complication.”

We thank the reviewers for their fair and very constructive feedback. We have done the appropriate modifications according to our experience and convictions. We are convinced that by the modifications done to the manuscript according to your suggestions we have highly improved the quality of our paper.

Round 2

Reviewer 1 Report

All edits incorporated or addressed, well done.

Reviewer 2 Report

I would like to thank the authors for responding to my comments. I believe that this papers is worth publishing as it is an interesting topic and few research is on the topic. 

Regards,